# Rethinking the Reverse-engineering of Trojan Triggers

**Zhenting Wang**
Rutgers University
zhenting.wang@rutgers.edu

**Kai Mei**
Rutgers University
kai.mei@rutgers.edu

**Hailun Ding**
Rutgers University
hailun.ding@rutgers.edu

**Juan Zhai**
Rutgers University
juan.zhai@rutgers.edu

**Shiqing Ma**
Rutgers University
sm2283@rutgers.edu

## Abstract

Deep Neural Networks are vulnerable to Trojan (or backdoor) attacks. Reverse-engineering methods can reconstruct the trigger and thus identify affected models. Existing reverse-engineering methods only consider input space constraints, e.g., trigger size in the input space. Expressly, they assume the triggers are static patterns in the input space and fail to detect models with feature space triggers such as image style transformations. We observe that both input-space and feature-space Trojans are associated with feature space hyperplanes. Based on this observation, we design a novel reverse-engineering method that exploits the feature space constraint to reverse-engineer Trojan triggers. Results on four datasets and seven different attacks demonstrate that our solution effectively defends both input-space and feature-space Trojans. It outperforms state-of-the-art reverse-engineering methods and other types of defenses in both Trojaned model detection and mitigation tasks. On average, the detection accuracy of our method is 93%. For Trojan mitigation, our method can reduce the ASR (attack success rate) to only 0.26% with the BA (benign accuracy) remaining nearly unchanged. Our code can be found at https://github.com/RU-System-Software-and-Security/FeatureRE.

## 1 Introduction

DNNs are vulnerable to Trojan attacks [1–6]. After injecting a Trojan into the DNN model, the adversary can manipulate the model prediction by adding a *Trojan trigger* to get the target label. The adversary can inject the Trojan by performing the poisoning attack or supply chain attack. In the poisoning attack, the adversary can control the training dataset and injects the Trojan by adding samples with the Trojan trigger labeled as the target label. In the supply chain attack, the adversary can replace a benign model with a Trojaned model by performing the supply chain attack. The Trojan trigger is becoming more and more stealthy. Earlier works use static patterns, e.g., a yellow pad as the trigger, which is known as the input space triggers. Researchers recently proposed using more dynamic and input-aware techniques to generate stealthy triggers that mix with benign features, which are referred to as the feature space triggers. For example, the trigger of the feature-space Trojans can be a warping process [7] or a generative model [3, 8, 9]. The Trojan attack is a prominent threat to the trustworthiness of DNN models, especially in security-critical applications, such as autonomous driving [1], malware classification [10], and face recognition [11].

Prior works have proposed several ways to defend against Trojan attacks, such as removing poisons in training [12–14], detecting Trojan samples at runtime [15–18], etc. Many of above methods can only work for one type of Trojan attack. For example, training and pre-training time defense (e.g., removing poisoning data, training a benign model with poisoning data) fail to defend against the supply chain attack. Trigger reverse-engineering [19–23] is a general method to defend against

different Trojan attacks under different threat models. It works by searching for if there exists an input pattern that can be used as a trigger in the given model. If we can find such a trigger, the model has a corresponding Trojan and is marked as malicious and vice versa. Existing reverse-engineering methods assume that the Trojan triggers are static patterns in the input space and develop an optimization problem that looks for an input pattern that can be used as the trigger. This assumption is valid for input space attacks [1, 2, 24] that use static triggers (e.g., a colored patch). Feature space attacks [3–5, 7–9, 25] break this assumption. Existing trigger reverse-engineering methods [19–23] constrain the optimization by using heuristics or empirical observations on existing attacks, such as pixel values are in range $[0, 255]$, and the trigger's size is small. Such heuristics are also invalid for feature space triggers that change all pixels in images. Reverse-engineering the feature space is challenging. Unlike input space, there are no constraints that can be directly used.

In this paper, we propose a trigger reverse-engineering method that works for feature space triggers. Our intuition is that *features representing the Trojan are orthogonal to other features.* Because a trigger works for a set of samples (or all of them, depending on the attack type), changing the input content without removing the Trojan features will not change the prediction. That is, changing Trojan and benign features will not affect each other. Trojan features will form a hyperplane in the high dimensional space, which can constrain the search in feature space. We then develop our

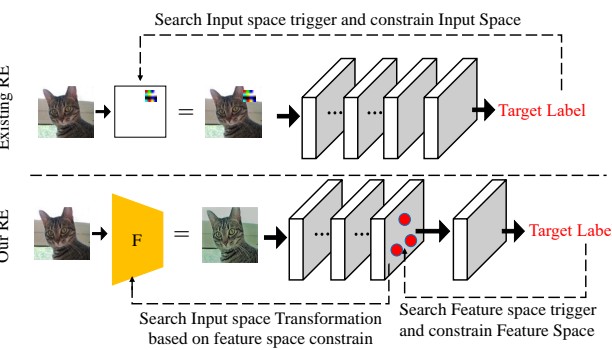

Fig. 1: Existing reverse-engineering (RE) and ours.

reverse-engineering method by exploiting the feature space constraint. Fig. 1 demonstrates our idea. Existing reverse-engineering methods only consider the input space constraint. It conducts reverse-engineering via searching a static trigger pattern in the input space. These methods fail to reverse-engineer feature-space Trojans whose trigger is dynamic in the input space. Instead, our idea is to exploit the feature space constraint and searching a feature space trigger using the constraint that the Trojan features will form a hyperplane. At the same time, we also reverse-engineer the input space Trojan transformation based on the feature space constraint. To the best of our knowledge, we are the first to propose feature space reverse-engineering methods for backdoor detection.

Through reverse-engineered Trojans, we developed a Trojan detection and removal method. We implemented a prototype FEATURERE (**FEATURE**-space **RE**verse-engineering) in Python and PyTorch and evaluated it on MNIST, GTSRB, CIFAR, and ImageNet dataset with seven different attacks (i.e., BadNets [1], Filter attack [20], WaNet [7], Input-aware dynamic attack [8], ISSBA [9], Clean-label attack [26], Label-specific attack [1], and SIG attack [27]). Our results show that FEATURERE is effective. On average, the detection accuracy of our method is 93%, outperforming existing techniques. For Trojan mitigation, our method can reduce the ASR (attack success rate) to only 0.26% with the BA (benign accuracy) remaining nearly unchanged by using only ten clean samples for each class.

Our contributions are summarized as follows. We first find the feature space properties of the Trojaned model and reveal the relationship between Trojans and the feature space hyperplanes. We propose a novel Trojan trigger reverse-engineering technique leveraging the feature space Trojan hyperplane. We evaluate our prototype on four different datasets, five different network architectures, and seven advanced Trojan attacks. Results show that our method outperforms SOTA approaches.

## 2 Background & Motivation

A DNN classifier is a function $\mathcal{M} : \mathcal{X} \mapsto \mathcal{Y}$ where $\mathcal{X}$ is the input domain $\mathbb{R}^m$ and $\mathcal{Y}$ is a set of labels $K$. A Trojan (or backdoor) attack against a DNN model $\mathcal{M}$ is a malicious way of perturbing the input so that an adversarial input $x'$ (i.e., input with the perturbation pattern) will be classified to a target/random label while the model maintains high accuracy for benign input $x$. The perturbation pattern is known as the Trojan trigger. Trojan attacks can happen in training (e.g., data poisoning)

or model distribution (e.g., changing model weights or supply-chain attack). Existing works have shown Trojan attacks against different DNN models, including computer vision models [1, 2, 24], Graph Neural Networks (GNNs) [28, 29], Reinforcement Learning (RL) [30, 31], Natural Language Processing (NLP) [32–37], recommendation systems [38], malware detection [10], pretrained models [21, 39, 40], active learning [41], and federated learning [42, 43]. The Trojan trigger can be a simple input pattern (e.g., a yellow pad) [1, 2, 24] or a complex input transformation function (e.g., a CycleGAN to change the input styles) [3, 5, 7–9]. If the trigger is static input space perturbations (e.g., a yellow pad), the Trojan attack is known as *input-space Trojan*, and if the trigger is an input feature (e.g., an image style), the attack is referred to as the *feature-space Trojan*.

There are different types of Trojan defenses. A line of work [13, 14, 44] attempts to remove poisoned data samples by cleaning the training dataset. Training-based methods [45–47] train benign classifiers even with the poisoned dataset. These training time approaches work for poisoning-based attacks but fail to defend against supply chain attacks where the adversary injects the Trojan after the model is trained. Another line of work, e.g., STRIP [15], SentiNet [16], and Februus [17] aim to detect Trojan inputs during runtime. It is hard to distinguish between a misclassification and a Trojan attack for a test input. These runtime detection methods make assumptions about the attack, which stronger attacks can violate. For example, STRIP fails to detect the Trojan inputs when the Trojan trigger locates around the center of an image or overlaps with the main object (e.g., feature space attacks). Another limitation is that they examine the test inputs and perform various heavyweight tests, significantly delaying the response time.

Trigger reverse engineering [19–23, 48–50] makes no assumptions about the attack method (e.g., poisoning or supply-chain attacks) and does not affect the runtime performance. It inspects the model to check if a Trojan exists before deploying. Given a DNN model $\mathcal{M}$ and a small set of clean samples $\mathcal{X}$, trigger reverse engineering methods try to reconstruct injected triggers. If reverse engineering is successful, the model is marked as malicious. Neural Cleanse (NC) [19] proposes to perform reverse engineering by solving Eq. 1:

$$\min_{\boldsymbol{m},\boldsymbol{t}} \quad \mathcal{L}\left(\mathcal{M}\left((1-\boldsymbol{m})\odot\boldsymbol{x}+\boldsymbol{m}\odot\boldsymbol{t}\right),y_t\right)+r^{\star} \tag{1}$$

where $x \in \mathcal{X}$ and $\boldsymbol{m}$ is the trigger mask (i.e., a binary matrix with the same size as the input to determine if the value will be replaced by the trigger or not), $\boldsymbol{t}$ is the trigger pattern (i.e., a matrix with the same size as the input containing trigger values), and $r^{\star}$ are attack constraints (e.g., trigger size is smaller than 1/4 of the image). $\mathcal{L}$ is the cross-entropy loss function. Most prior works [20–23] follow the same methodology and inherently suffer from the same limitations. First, they assume that an input space perturbation, denoted by $(\boldsymbol{m}, \boldsymbol{t})$, can represent a trigger. This assumption is valid for input-space triggers but does not hold for feature space attacks. Second, $r^{\star}$ are heuristics observed from existing attacks. For example, NC observed that most triggers have small sizes and limit the trigger size to be no larger than a threshold value. Otherwise, the trigger will overlap with the main object and decrease benign accuracy. In practice, more advanced attacks can break such heuristics. For instance, DFST [3] leverages CycleGAN to transfer images from one style to another without changing its semantics. It changes almost all pixels in a given image. This paper proposes a novel reverse engineering method that overcomes the limitations above for image classifiers.

## 3 Methodology

### 3.1 Threat Model

This work aims to determine if a given model has a Trojan or not by reverse-engineering the corresponding trigger. Following existing works [19, 20, 51], we assume access to the model and a small dataset containing correctly labeled benign samples of each label. In practice, such datasets can be gathered from the Internet. We make no assumptions on how the attacker injects the Trojan (poisoning or supply-chain attack). The attack can be formally defined as: $\mathcal{M}(\boldsymbol{x}) = y, \mathcal{M}(F(\boldsymbol{x})) = y_T, \boldsymbol{x} \in \mathcal{X}$, where $\mathcal{M}$ is the Trojaned model, $\boldsymbol{x}$ is a clean input sample, and $y_T$ is the target label. $F$ is the function to construct Trojan samples. Input-space triggers add static input perturbations, and feature space triggers are input transformations. The key difference between our work and existing work is that we consider the feature space triggers.

## 3.2 Observation

In DNNs, the neuron activation values represent its functionality. The input neurons denote the input space features, and inner neurons extract inner and more abstract features. Existing reverse-engineering methods constrain the optimization problem in the input space using domain-specific constraints or observations. For image classification tasks, the pixel value of each image has to be a valid RGB value. Methods like NC observe that the trigger size must be smaller and cannot overlap with the main object and propose corresponding constraints. The most challenging problem for reverse-engineering feature space triggers is how to constrain the optimization properly. Note that there exist a set of neurons; when activating to specific values, the Trojan behavior will be triggered. Due to the black-box nature of DNNs, it is hard to identify which neurons are related to the Trojan behavior. Moreover, if enlarge the weight values with the same scale, the output of the DNN will be the same, and as such, it is hard to constrain concrete activation values. Without a proper constraint, we cannot form an optimization problem.

Our key observation to solve this problem is that *neuron activation values representing the Trojan behavior are orthogonal to others*. Recall that one property of DNN Trojans is that when adding the trigger to *any* given input, the model will predict the output to a specific label. That is, the trigger will always work regardless of the actual contents of the input. In the feature space, when the model recognizes features of the Trojan, it will predict the label to the target label regardless of the other features. These activation values will form a hyperplane space in the high dimensional space so that they can be orthogonal to all others. Based on this intuition, we performed empirical experiments to confirm our idea. Specifically, we first use six Trojan attacks (e.g., BadNets [1], Clean label attack [26], Filter attack [20], and WaNet [7], SIG [27] and Input-aware dynamic attack [8]) to generate Trojaned ResNet18 models on CIFAR-10. We then visualize the feature space of the last convolutional layers in these models. In Fig. 2, three dimensions, X, Y, and Z, represent the feature space. We first apply PCA to get two eigenvectors of the benign training set; then, we use the obtained eigenvectors as X-axis and Y-axis. For Z-axis, we first construct Trojan inputs to activate the model's Trojan behavior and find highly related neurons to Trojans. Then, we use DNN interpretability techniques SHAP [52] to estimate the neuron's importance to the Trojan behavior. The neurons among the top 3% are *compromised neurons*. Z-axis denotes the activation values of compromised neurons. Namely, $z = \|\mathcal{A}(F(\boldsymbol{x})) \odot \boldsymbol{m}\|$ , where $\boldsymbol{m}$ denotes a mask revealing the position of compromised neurons. Fig. 2 show that most Trojan inputs have a similar z-value. They form a linear hyperplane in the feature space while benign ones do not.

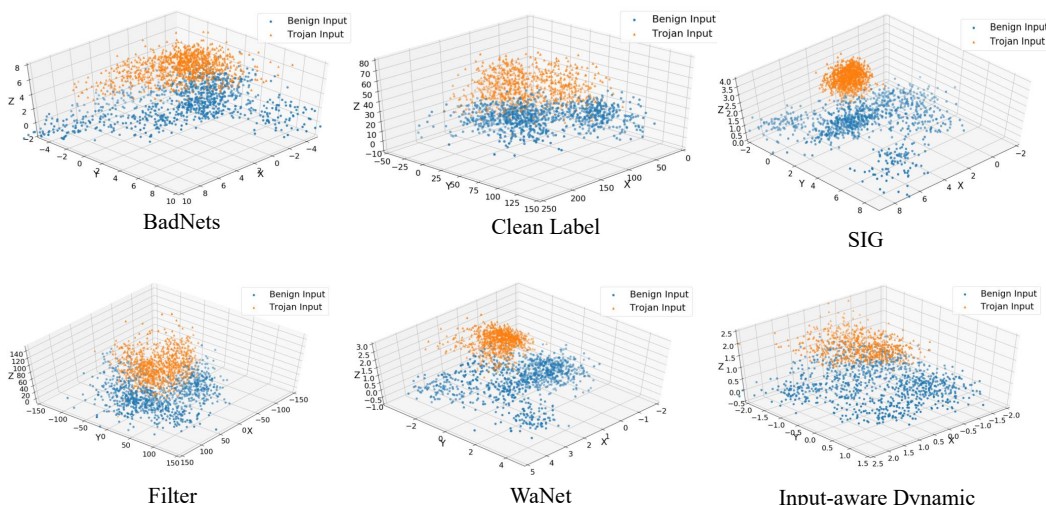

Fig. 2: Feature space of Trojaned models.

## 3.3 Feature Space Trojan Hyperplane Reverse-engineering

In this paper, We use $\mathcal{A}$ to represent the submodel from the input space to the feature space. $\mathcal{B}$ is the submodel from the feature space to the output space. We also use $\boldsymbol{a} = \mathcal{A}(\boldsymbol{x})$ to denote the inner

---
**Algorithm 1** Feature-space Backdoor Reverse-engineering
---
**Input:** Model: $\mathcal{M}$
**Output:** Trojaned or Not, Trojaned Pairs $T$
 1: **function** REVERSE-ENGINEERING($\mathcal{M}$)
 2:     **for** (target class $y_t$, source class $y_s$ ) in $K$ **do**
 3:         **for** $e \leq E$ **do**
 4:             $\boldsymbol{x} = sample(\mathcal{X}_{y_s})$
 5:             $cost_1 = \mathcal{L}\left(\mathcal{B}\left((1-\boldsymbol{m})\odot\boldsymbol{a} + \boldsymbol{m}\odot\boldsymbol{t}\right), y_t\right)$
 6:             **if** $\|F(\boldsymbol{x}) - \boldsymbol{x}\| \geq \tau_1$ **then**
 7:                $cost_1 = cost_1 + w_1 \cdot \|F(\boldsymbol{x}) - \boldsymbol{x}\|$
 8:             **if** $std(\boldsymbol{m}\odot\mathcal{A}(F(\boldsymbol{x}))) \geq \tau_2$ **then**
 9:                $cost_1 = cost_1 + w_2 \cdot std(\boldsymbol{m}\odot\mathcal{A}(F(\boldsymbol{x})))$
10:             $\Delta_{\theta_F} = \frac{\partial cost_1}{\partial \theta_F}$
11:             $\theta_F = \theta_F - lr_1 \cdot \Delta_{\theta_F}$
12:             $cost_2 = \mathcal{L}\left(\mathcal{B}\left((1-\boldsymbol{m})\odot\boldsymbol{a} + \boldsymbol{m}\odot\boldsymbol{t}\right), y_t\right)$
13:             **if** $\|\boldsymbol{m}\| \geq \tau_3$ **then**
14:                $cost_2 = cost_2 + w_3 \cdot \|\boldsymbol{m}\|$
15:             $\Delta_{\boldsymbol{m}} = \frac{\partial cost_2}{\partial \boldsymbol{m}}$
16:             $\boldsymbol{m} = \boldsymbol{m} - lr_2 \cdot \Delta_{\boldsymbol{m}}$
17:         **if** $ASR\left(\mathcal{B}\left((1-\boldsymbol{m})\odot\boldsymbol{a} + \boldsymbol{m}\odot\boldsymbol{t}\right), y_t\right) > \lambda$ **then**
18:             $\mathcal{M}$ is a Trojaned model,
19:             $T.append((y_s, y_t))$
---

features of the model. Similar to the reverse-engineering in the input space, given a model $\mathcal{M}$ and a small set of benign inputs $\mathcal{X}$, we use a feature space mask $\boldsymbol{m}$ and a feature space pattern $\boldsymbol{t}$ to represent the feature space Trojan hyperplane $H = \{\boldsymbol{a}|\boldsymbol{m}\odot\boldsymbol{a} = \boldsymbol{m}\odot\boldsymbol{t}\}$. Specifically, we can update $\boldsymbol{m}$ and $\boldsymbol{t}$ via the following optimization process: $\min_{\boldsymbol{m},\boldsymbol{t}}\mathcal{L}\left(\mathcal{B}\left((1-\boldsymbol{m})\odot\boldsymbol{a} + \boldsymbol{m}\odot\boldsymbol{t}\right), y_t\right)$. $y_t$ is the target label. As discussed above, reverse-engineering the feature space is challenging. In the input space, all values have natural physical semantics and constraints, e.g., a pixel value in the RGB value range. Values in the feature space have uninterruptible meanings and are not strictly constrained. Whether the result will have a physically meaningful semantic is also uncertain. We solve these challenges by simultaneously optimizing the input space trigger function $F$ and the feature space Trojan hyperplane $H$ to enforce that the trigger has semantic meanings. In detail, we compute the feature space trigger pattern as the mean inner features on the samples generated by the trigger function, i.e., $\boldsymbol{t} = mean\left(\boldsymbol{m}\odot\mathcal{A}(F(\mathcal{X}))\right)$. We also constrain the standard deviation of $\boldsymbol{m}\odot\mathcal{A}(F(\mathcal{X}))$ to make sure the features generated by the trigger function will lie on the relaxation of the reverse-engineered hyperplane. Formally, our reverse-engineering can be written as the constrained optimization problem shown in Eq. 2, where $\mathcal{X}$ is the small set of clean samples. We use deep neural networks to model the trigger function (i.e., $F = G_\theta$) because of their expressiveness [23, 53]. Specifically, we use a representative deep neural network UNet [54]. Given a model and a small set of clean inputs, the trigger function can be smoothly reconstructed via gradient-based methods, i.e., optimizing the generative model $G_\theta$. In our default setting, $\mathcal{A}$ and $\mathcal{B}$ are separated at the last convolutional layer. More discussions are in the Appendix (§ 4.5).

$$\min_{F,\boldsymbol{m}} \mathcal{L}\left(\mathcal{B}\left((1-\boldsymbol{m})\odot\boldsymbol{a} + \boldsymbol{m}\odot\boldsymbol{t}\right), y_t\right)$$
$$\text{where } \boldsymbol{t} = mean\left(\boldsymbol{m}\odot\mathcal{A}(F(\mathcal{X}))\right), \boldsymbol{a}\in\mathcal{A}(\mathcal{X}) \tag{2}$$
$$s.t. \quad \|F(\mathcal{X}) - \mathcal{X}\| \leq \tau_1, \; std(\boldsymbol{m}\odot\mathcal{A}(F(\mathcal{X}))) \leq \tau_2, \; \|\boldsymbol{m}\| \leq \tau_3$$

There are several constraints in the optimization problem: ① The transformed samples should be similar to the original image due to the properties of Trojan attacks, i.e., $\|F(\boldsymbol{x}) - \boldsymbol{x}\| \leq \tau_1$. Typically, the Trojan samples are visually similar to original samples for stealthy purposes. In detail, we use MSE (Mean Squared Error) to calculate the distance between $F(\boldsymbol{x})$ and $\boldsymbol{x}$. ② The Trojan features should lie in the relaxation of the reverse-engineered feature space Trojan hyperplane, i.e., $\mathbb{P}\left(a\in H^\star | \boldsymbol{x}\in F(\mathcal{X})\right)$ should have high values. To achieve this goal, we constrain the standard

deviation of different Trojan samples' activation values on each pixel in the hyperplane. ③ Similar to input space trigger reverse-engineering [20], we set a bound for the size of the feature space trigger mask, i.e., $\|\boldsymbol{m}\| \leq \tau_3$. Here $\tau_1$, $\tau_2$, and $\tau_3$ are threshold values. We discuss their influence in § 4.4. The detailed reverse-engineering algorithm can be found in Algorithm 1, where $K$ is a set containing all possible (source class, target class) pairs of the model. FEATURERE scans all labels to identify the Trojan target labels. $w_1$, $w_2$ and $w_3$ are the coefficient values used in the optimization. Following NC [19], we adjust them dynamically during optimization to make sure the reverse-engineered Trojan satisfies the constraints. $E$ is the maximal epoch number. In lines 5-11, it optimizes the trigger function $F$ and then the mask $\boldsymbol{m}$ of the feature space hyperplane in lines 12-16. In the end, we determine the reverse-engineering is successful and the label $y_t$ is a Trojan target label if the attack success rate of the reversed Trojan is above a threshold value $\lambda$ (80% in this paper).

### 3.4 Trojan Mitigation

After we reverse-engineered the Trojans, we can mitigate it by breaking the reverse-engineered feature space Trojan hyperplane. Based on our observation, the neurons in the feature space Trojan hyperplane are highly related to the Trojan behaviors. Thus, we can mitigate the Trojans by breaking the hyperplane. Inspired by Zhao et al. [55], we can break it by flipping the neurons on it. Our neuron-flip process can be written as Eq. 3, where $\boldsymbol{m}$ is the reverse-engineered feature space mask, $\boldsymbol{a}$ is the inner features. $\boldsymbol{a}_i$ is the activation value on the $i^{th}$ neuron.

$$Flip(\boldsymbol{a}) = \begin{cases} -\boldsymbol{a}_i, & \text{when } \boldsymbol{a}_i \text{ in } \boldsymbol{m} \\ \boldsymbol{a}_i, & \text{when } \boldsymbol{a}_i \text{ not in } \boldsymbol{m} \end{cases} \tag{3}$$

The mitigated model $\mathcal{M}'(\boldsymbol{x}) = \mathcal{B}\left(Flip(\mathcal{A}(\boldsymbol{x}))\right)$, where $\mathcal{A}$ and $\mathcal{B}$ are submodels of the model.

## 4 Experiments and Results

We first introduce our experiment setup (§ 4.1). We then evaluate the effectiveness of FEATURERE on Trojan detection (§ 4.2) and mitigation tasks (§ 4.3). We also evaluate the robustness of FEATURERE against different settings and the impacts of configurable parameters in FEATURERE (§ 4.4). In § 4.5, we discuss how to split the model. The adaptive attack can be found in § 4.6.

### 4.1 Experiment Setup.

We implement FEATURERE with python 3.8 and PyTorch. All experiments are conducted on a Ubuntu 18.04 machine equipped with 64 CPUs and six GeForce RTX 6000 GPUs.

**Datasets and Models.** We use four publicly available datasets to evaluate FEATURERE, including MNIST [56], GTSRB [57], CIFAR-10 [58] and ImageNet [59]. We summarize our datasets in Table 1. We show the dataset names, the size of each input sample, the number of samples and the number of classes in each column. Details of the datasets can be found in Appendix. For model architectures, we use LeNet5 [56], Preact ResNet18 (PRN18) [60], ResNet18 [61], a VGG-style network spec-

Table 1: Overview of datasets.

| Dataset | Sample Size | #Train | Classes |
|---|---|---|---|
| MNIST | 32*32*1 | 60000 | 10 |
| GTSRB | 32*32*3 | 39209 | 43 |
| CIFAR-10 | 32*32*3 | 50000 | 10 |
| ImageNet | 224*224*3 | 100000 | 200 |

ified in ULP [62], and a model consists of 4 convolutional layers and 2 dense layers used in Xu et al. [51]. These datasets and models are widely used in Trojan-related researches [1–3, 15, 19, 20, 51, 63].

**Evaluation Metrics.** We measure the effectiveness of the Trojan detection task by collecting the detection accuracy (Acc). Given a set of models consist of benign and Trojaned models, the Acc is the number of correctly classified models over the number of all models. We also show detailed number of True Positives (TP, i.e., correctly detected Trojaned models), False Positives (FP, i.e., benign models classified as Trojaned models), False Negatives (FN, i.e., Trojaned models classified as benign models) and True Negatives (TN, i.e., correctly classified benign models). For the Trojan mitigation task, we evaluate the benign accuracy (BA) and attack success rate (ASR) [64]. BA is the number of correctly classified clean inputs over the number of all clean samples. ASR is defined as the number of Trojan samples that successfully attack models over the number of all Trojan samples.

Table 2: Comparison to reverse-engineering methods.

| Dataset | Network | Attack | ABS | | | | | DeepInspect | | | | | TABOR | | | | | K-arm | | | | | FeatureRE | | | | |
|---------|---------|--------|----|----|----|----|-----|----|----|----|----|-----|----|----|----|----|-----|----|----|----|----|-----|----|----|----|----|-----|
| | | | TP | FP | FN | TN | Acc | TP | FP | FN | TN | Acc | TP | FP | FN | TN | Acc | TP | FP | FN | TN | Acc | TP | FP | FN | TN | Acc |
| MNIST | LeNet5 | WaNet | 7 | 2 | 3 | 8 | 75% | 4 | 0 | 6 | 10 | 70% | 3 | 2 | 7 | 8 | 55% | 5 | 0 | 5 | 10 | 75% | 9 | 1 | 1 | 9 | **90%** |
| GTSRB | PRN18 | WaNet | 5 | 0 | 5 | 10 | 75% | 5 | 1 | 5 | 9 | 70% | 2 | 2 | 8 | 8 | 50% | 4 | 0 | 6 | 10 | 70% | 8 | 0 | 2 | 10 | **90%** |
| CIFAR-10 | ResNet18 | BadNets | 18 | 0 | 2 | 20 | 95% | 20 | 2 | 0 | 18 | 95% | 20 | 3 | 0 | 17 | 93% | 20 | 0 | 0 | 20 | **100%** | 20 | 1 | 0 | 19 | 98% |
| | | Filter | 13 | 0 | 7 | 20 | 83% | 6 | 2 | 14 | 18 | 60% | 5 | 3 | 15 | 17 | 55% | 0 | 0 | 20 | 20 | 50% | 18 | 1 | 2 | 19 | **93%** |
| | | WaNet | 11 | 0 | 9 | 20 | 78% | 11 | 2 | 9 | 18 | 73% | 3 | 3 | 17 | 17 | 50% | 9 | 0 | 11 | 20 | 73% | 18 | 1 | 2 | 19 | **93%** |
| | | IA | 3 | 0 | 17 | 20 | 58% | 4 | 2 | 16 | 18 | 55% | 3 | 3 | 17 | 17 | 50% | 2 | 0 | 18 | 20 | 55% | 19 | 1 | 1 | 19 | **95%** |

Table 3: Comparison to ULP.

| Network | Attack | ULP | | | | | FeatureRE | | | | |
|---------|--------|-----|----|----|----|-----|----|----|----|----|-----|
| | | TP | FP | FN | TN | Acc | TP | FP | FN | TN | Acc |
| VGG | WaNet | 1 | 0 | 19 | 20 | 53% | 17 | 0 | 3 | 20 | **93%** |

Table 4: Comparison to Meta-classifier.

| Network | Attack | Meta Classifier | | | | | FeatureRE | | | | |
|---------|--------|-----|----|----|----|-----|----|----|----|----|-----|
| | | TP | FP | FN | TN | Acc | TP | FP | FN | TN | Acc |
| 4Conv+2FC | WaNet | 16 | 4 | 4 | 16 | 80% | 18 | 0 | 2 | 20 | **95%** |

**Baselines and Attack Settings.** We evaluate the performance of FEATURERE on Trojan detection, and compare the results with four reverse-engineering based Trojan detection methods (i.e., ABS [20], DeepInspect [23], TABOR [22], and K-arm [21]) and two classification based methods (i.e., ULP [62] and Meta-classifier [51]). For Trojan mitigation task, we compare FEATURERE with two advanced mitigation methods (i.e., NAD [65] and I-BAU [66]). We use the default parameter settings described in the original papers of our baseline methods. To understand the performance of FEATURERE and existing methods against various attack settings, we evaluate them against BadNets [1], Filter Trojans [20], WaNets [7], IA (Input-dependent dynamic Trojans) [8], Clean-label [26], SIG [27] and ISSBA (Invisible sample-specific Trojans) [9] attacks. These attacks are state-of-the-art attack methods and are widely evaluated in Trojan defense papers [19, 20, 66, 45]. If not specified, we use all-to-one (i.e., single-target) setting for all attacks. Label-specific setting is discussed in § 4.4.

## 4.2 Effectiveness on Trojan Detection

To measure the effectiveness on the Trojan detection task, we generate a set of benign and Trojaned models, and then use FEATURERE and existing Trojan detection methods to classify each model. We collect the Acc, TP, FP, FN and TN results of each method and compare them. Specifically, we first evaluate the performance of FEATURERE and compare the results with four state-of-the-art reverse-engineering based detection methods. We generate 20 Trojaned models as well as 20 benign models on CIFAR-10 dataset for each attack (i.e., BadNets, Filter Trojan, WaNet and Input-aware dynamic Trojan attack). For MNIST and GTSRB dataset, we train 10 Trojaned and 10 benign LeNet5 [56] models on each dataset. We then compare FEATURERE with two state-of-the-art classification based detection methods. Similarly, we generate 10 benign and 10 Trojaned models, and use Trojan detection methods to classify these models. Notice that, in all Trojan detection tasks, we assume the defender can only access 10 clean samples for each class, which is a common practice. [19–21] The comparison results of reverse-engineering based methods are shown in Table 2. The results of two classification based methods are demonstrated in Table 3 and Table 4. In each table, we show the detailed settings, including dataset names, network architectures, and attack settings.

**Comparison to Reverse-engineering based methods.** From the results in Table 2, we observe that FEATURERE achieves the best detection results compared with other methods. The average Acc of FEATURERE is 93%, which is 17%, 23%, 35% and 23% higher than those of other defense methods. The results show the benefit of FEATURERE. When looking into the generalization of Trojan detection methods, we find that FEATURERE can achieve excellent results on both input-space Trojans (i.e., BadNets) and feature-space Trojans (i.e., Filter, WaNet and IA attacks). However, the performance of existing reverse-engineering methods on feature-space Trojans (i.e., Filter, WaNet and IA attacks) is significantly worse than the performance on static Trojans. FEATURERE archives 94% average Acc but the Acc of TABOR on feature-space Trojans are only 53%, 50% and 50%, respectively. Moreover, FEATURERE has 15.33 TP on average, but existing methods only have 7.87 TP. FEATURERE can generalize better than existing work because FEATURERE considers both feature and input space constraints. Existing methods, on the contrary, only consider the input space

Table 5: Results on Trojan mitigation task (10 clean samples for each class are used).

| Dataset | Network | Attack | Undefended | | NAD | | I-BAU | | FeatureRE | |
|---|---|---|---|---|---|---|---|---|---|---|
| | | | BA | ASR | BA | ASR | BA | ASR | BA | ASR |
| MNIST | LeNet5 | WaNet | 99.22% | 94.52% | 65.87% | 48.21% | 94.87% | **0.22%** | 99.20% | 0.63% |
| GTSRB | PRN18 | WaNet | 99.02% | 99.70% | 70.35% | 62.76% | 91.74% | 0.86% | 98.42% | **0.00%** |
| CIFAR-10 | ResNet18 | Filter | 91.30% | 98.98% | 81.66% | 28.23% | 87.45% | 18.22% | 91.26% | **0.29%** |
| | | WaNet | 91.84% | 98.17% | 83.60% | 27.52% | 87.52% | 6.84% | 91.79% | **0.04%** |
| | | IA | 91.62% | 92.44% | 84.03% | 34.00% | 86.88% | 10.33% | 91.43% | **0.38%** |

constraints. They can not detect feature-space Trojans whose trigger is complex and input-dependent, and directly classify many Trojaned models with feature-space Trojans as benign.

**Comparison to classification based methods.** When comparing FEATURERE with classification based methods, we notice that FEATURERE has better Acc, more TPs and TNs than classification based methods ULP and Meta-classifier. As demonstrated in Table 3 and Table 4, the Acc of FEATURERE is 93% and 95%, which is 40% and 15% higher than those of ULP and Meta-classifier. Overall, the results indicate that FEATURERE is more effective than classification based methods when detecting Trojaned models. Different from FEATURERE, which directly inspects models via analyzing its inherent feature space properties, classification based methods highly depend on the external trained dataset. Therefore, their results are not as precise as FEATURERE.

## 4.3 Effectiveness on Trojan Mitigation

We evaluate the effectiveness of FEATURERE on Trojan mitigation and compare the results with state-of-the-art methods NAD and I-BAU. We use the Trojaned models generated by three attacks (i.e., Filter attack, WaNet and IA) and report their average BA and ASR after Trojan mitigation. We also show the average BA and ASR of undefended Trojaned models. For all methods, the defenders can access 10 clean samples for each class to conduct Trojan mitigation. We show the results in Table 5.

We find that FEATURERE is the most effective method for Trojan mitigation among all methods. Compared to state-of-the-art Trojan mitigation methods, FEATURERE archives the lowest average ASR and the highest average BA. On the one hand, using FEATURERE can decrease the average ASR from 96.76% to 0.26%. Other methods can only decrease the average ASR to 40.14% and 7.29%. The results show the advantages of FEATURERE on Trojan mitigation. On the other hand, the BA with FEATURERE is similar to undefended models. But the BA of other methods is significantly lower than that of undefended models. By breaking the feature space hyperplane, FEATURERE can successfully mitigate Trojans with minimal BA loss. Other methods, which cannot effectively find Trojan-related features, cannot achieve good results.

## 4.4 Ablation Study

In this section, we evaluate the resistance of FEATURERE to various Trojan attack settings and large datasets. We also evaluate the impacts of configurable parameters in FEATURERE, including the constrain values used in Eq. 2 and the number of used clean samples. By default, the attack used for measuring the impacts of configurable parameters is IA. We use 20 benign ResNet models and 20 Trojaned ResNet models on CIFAR-10 to test the detection results. Notice that we only evaluate the performance on the Trojan detection task. Due to the page limits, we include the ablation study on Trojan mitigation in Appendix (§ A.4).

**Resistance to various attack and dataset settings.**
To evaluate if our method is resistant to more Trojan attacks, we train 20 Trojaned ResNet18 models on CIFAR-10 for Label-specific attack (LS), Clean-label attack (CL) and SIG attack (SIG). For the label-specific attack, we consider the all-to-all attack setting, i.e., the target label $y_T = \eta(y) = y + 1$, where $\eta$ is a mapping and $y$ is the correct label of the sample. In addition, we generate five benign models and five

Table 6: Resistance to more attacks.

| Dataset | Network | Attack | TP | FP | FN | TN | Acc |
|---|---|---|---|---|---|---|---|
| CIFAR-10 | ResNet18 | LS | 9 | 1 | 1 | 9 | 90% |
| | | CL | 8 | 1 | 2 | 9 | 85% |
| | | SIG | 10 | 1 | 0 | 9 | 95% |
| ImageNet | ResNet18 | ISSBA | 4 | 0 | 1 | 5 | 90% |

Trojaned models with ISSBA attacks on ImageNet to evaluate if our method is compatible with large-scale datasets. We summarize the results in Table 6.

In Table 6, we find that FEATURERE is compatible with evaluated Trojan attacks, showing the generalization of our reverse-engineering based method. We also observe that our method has high Acc on the ImageNet dataset with ISSBA [9]. Thus, our method is also applicable to large datasets.

**Influence of constrain values.** As shown in Eq. 2, there are three constrain values ($\tau_1$, $\tau_2$, $\tau_3$) in our constrained optimization process. By default, $\tau_1 = 0.15$, $\tau_2 = 0.25$ and $\tau_3 = 5\%$. We evaluate their influences. For $\tau_1$, we calculate input space perturbations on the preprocessed inputs, and the details of the preprocessing can be found in Appendix (§ A.2). We vary $\tau_1$ from 0.05 to 0.35, change $\tau_2$ from 0.10 to 0.50, and tune $\tau_3$ from 3% of the whole feature space to 10% of the whole feature space. The results under different hyperparameter settings are shown in Table 7.

From the results, we observe that the performance of FEATURERE is insensitive to these three hyperparameters. In detail, when we vary $\tau_1$, $\tau_2$ and $\tau_3$, the Acc is stable. In all cases, our method always achieves over 90% detection accuracy. The results further show the robustness of FEATURERE. We also find that, when the value of all hyperparameters becomes lower, FEATURERE has more FN. On the contary, when its value is larger, more FP will be produced. This is understandable because lower constrain values mean a stricter criterion for a successful reverse-engineering.

Table 7: Influence of hyperparameters.

| Metric | $\tau_1$ | | | $\tau_2$ | | | $\tau_3$ | | |
|---|---|---|---|---|---|---|---|---|---|
| | 0.05 | 0.15 | 0.35 | 0.10 | 0.25 | 0.50 | 3% | 5% | 10% |
| TP | 18 | 19 | 20 | 17 | 19 | 19 | 17 | 19 | 19 |
| FP | 0 | 1 | 3 | 1 | 1 | 2 | 0 | 1 | 2 |
| FN | 2 | 1 | 0 | 3 | 1 | 1 | 3 | 1 | 1 |
| TN | 20 | 19 | 17 | 19 | 19 | 18 | 20 | 19 | 18 |
| Acc | 0.95 | 0.95 | 0.93 | 0.90 | 0.95 | 0.93 | 0.93 | 0.95 | 0.93 |

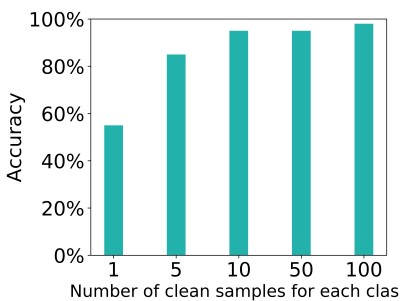

Fig. 3: Effects of clean set size.

**Number of clean reference samples.** Our threat model and existing work assume the defender can access a set of clean samples for defense. To investigate the influences of the number of used clean samples in Trojan detection, we choose the number from 1 to 100 in each class and report the Acc results. The results are shown in Fig. 3.

From the results, we notice that the Acc decreases significantly when we use less than 10 samples for each class. This is because the number of used sample affects the optimization process. When the number of used samples is too small, the optimization process might be problematic, e.g., it encounters overfitting problem. When the number of used samples is larger than 10, FEATURERE achieves high detection accuracy (i.e., above 95%) and the Acc will not change significantly when the number of used samples keeps increasing. The reason is using more data makes the optimizing process converge and finally arrives a stable state. Note that requiring hundreds clean samples is common for reverse-engineering based methods [19, 20, 22] and other types of defenses [15, 65, 66, 45]. FEATURERE only requires 10 clean samples for each class, which is more efficient.

### 4.5 Discussion for Model Split

As we discussed in § 3, our method split the model $\mathcal{M}$ into two sub-models $\mathcal{A}$ and $\mathcal{B}$. In this section, we discuss the influence of using different split positions. Table 8 shows the results of using different $\mathcal{A}$ and $\mathcal{B}$ on the ResNet18 model and CIFAR-10 dataset. In detail, we report the results of splitting the model at the 9th, 11th, 13th, 15th, and the last convolutional layer. The average detection accuracy for splitting at the 9th layer, 11th layer, 13th layer, 15th layer, and last layer is 86.50%, 87.75%, 89.50%, 94.00%, and 94.75%, respectively. As we can see, the performance of splitting at later layers is higher than the performance of splitting at earlier layers. In our current implementation, we set $A(x)$ as the sub-model from the input layer to the last convolution layer and $B(x)$ as the rest. The relationship between the input and the output of a convolutional layer $L_n$ is $x_{n+1} = L_n(x_n) = \sigma(\mathbf{W}_n^{\mathbf{T}} x_n + \mathbf{b}_n^{\mathbf{T}})$, where $x_n$ and $x_{n+1}$ are the inputs and outputs of layer $n$, $\mathbf{W}_n$ and $\mathbf{b}_n$ are weights and bias values, and $\sigma$ is the activation function.

Table 8: Accuracy on different split position.

| Attack | 9th | 11th | 13th | 15th | Last |
|---|---|---|---|---|---|
| BadNets | 88% | 93% | 98% | 98% | 98% |
| Filter | 88% | 85% | 90% | 95% | 93% |
| WaNet | 85% | 88% | 85% | 93% | 93% |
| IA | 85% | 85% | 85% | 90% | 95% |

Table 9: Results on BadNets and adaptive attack.

| Attack | BA | ASR | Acc |
|---|---|---|---|
| BadNets | 94.34% | 99.98% | 98% |
| Adaptive | 87.36% | 93.67% | 65% |

Based on existing literatures [67, 68], the features in the deeper CNN layers are more disentangled than that of earlier layers. Thus, if the orthogonal phenomenon happens in a layer $L_n$, it will exist for all its subsequent layers, e.g., $L_{n+1}$. If the orthogonal phenomenon does not happen, the layer without this phenomenon will mix benign and backdoor features, leading to low benign accuracy or attack success rate. The results in Table 9 confirm our analysis. Thus, a successful backdoor attack will lead to the orthogonal phenomenon in the last convolution layer.

### 4.6 Adaptive Attack

Our threat model assumes that the attacker can control the training process of the Trojan model. In this section, we discuss the potential adaptive attacker that knows our defense strategy and tries to bypass FEATURERE via modifying the training process. Our observation is that the neuron activation values representing the Trojan behavior are orthogonal to others. One possible adaptive attack is breaking such orthogonal relationships during the Trojan injection process. We design an adaptive attack that adds one loss term to push the Trojan features to be not orthogonal to benign features. This attack can be formulated as: $L = L_{ce} + L_{adv}$, where $L_{ce}$ is the standard classification loss and the $L_{adv}$ is defined as:

$$L_{adv} = \text{SIM}(\mathcal{B}(m \odot a + (1 - m) \odot t), \mathcal{B}(m \odot a' + (1 - m) \odot t)) \tag{4}$$

Here, SIM is the cosine similarity; $a$ and $a'$ are the features of different benign samples; $m$ and $t$ are the feature-space mask and pattern of the compromised neurons obtained via SHAP [52]. The loss term $L_{adv}$ tries to enforce the Trojan features being not orthogonal to the benign ones. We conduct this adaptive attack on the CIFAR-10 dataset and ResNet18 model. The results can be found in Table 9. The detection accuracy of FEATURERE under adaptive attack drops to 65%. Meanwhile, the average BA/ASR of the adaptive attack and BadNets (native training) is 87.36%/94.34% and 93.67%/99.98%, respectively. The adaptive attack can reduce the detection accuracy of our method. Both the BA and ASR of the adaptive attack are lower than those of native training. The results confirm our analysis in § 4.5: the model without the "orthogonal phenomenon" will mix benign and Trojan features, leading to low benign accuracy or attack success rate.

## 5 Discussion

**Limitations of our method.** Similar to most existing Trojaned model detection and mitigation methods [19, 20, 49, 21, 50, 22, 23, 63, 65, 48], our method requires a small set of clean samples. In the real world, these samples can be obtained from the Internet.

**Ethics.** This paper proposes a technique to detect and remove Trojans in DNN models. We believe it will help improve the security of DNNs and be beneficial to society.

## 6 Conclusion

In this paper, we find relationships between feature space hyperplane and Trojans in DNNs. More over, we propose a new Trojaned DNN detection and mitigation method based on our findings. Compared to the state-of-the-art methods, our method has better performance in both detection and mitigation tasks.

## Acknowledgement

We thank the anonymous reviewers for their constructive comments. This work is supported by IARPA TrojAI W911NF-19-S-0012. Any opinions, findings, and conclusions expressed in this paper are those of the authors only and do not necessarily reflect the views of any funding agencies.

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
