# A  Appendix

**Roadmap:** More details of Algorithm 1 is introduced in § A.1. Then, we present more details of the datasets (§ A.2) and attacks (§ A.3) used in the experiments. We also perform an ablation study for the Trojan mitigation task in § A.4. In § A.5, we visualize our reverse-engineered Trojans. We also show the generalization (§ A.6) and the efficiency (§ A.7) of FEATURERE. Finally, we discuss the findings in the evaluation (§ A.8).

## A.1  More details of Algorithm 1

In this section, we discuss more details of our Reverse-engineering Algorithm (Algorithm 1). Given a model $\mathcal{M}$ and a small set of clean samples $\mathcal{X}$, the output of the algorithm is a flag indicating if the model is Trojaned, and Trojaned label pairs denoting the source label and the target label of the detected Trojans.

In line 2, we iterate (source label, target label) pair from possible pairs $K$. $E$ in line 3 means the maximal optimization epoch number for each pair. It is set to 400 in this paper. In line 4, we randomly sample a batch of inputs from the samples in source classes. The batch size is set to 128 by default.

In lines 5 to Line 11, we optimize the parameters of the input space transformation $F$, which is represented by a UNet [54] model in our implementation. In line 5, we calculate the loss value specified in Eq. 1, where $a = \mathcal{A}(x)$ is the inner feature on clean samples. By default, $\mathcal{A}$ is the submodel from the input layer to the penultimate layer, and $\mathcal{B}$ is the submodel from the penultimate layer to the output layer. $m$ is the feature space trigger mask. $t = mean\,(m \odot \mathcal{A}(F(\mathcal{X}))$ is the feature space trigger pattern. $\mathcal{L}$ is the cross-entropy loss calculating the distance between the target label and the output of the model under inner features with feature space Trojans. In line 6, if the input space MSE (Mean Square Error) distance for original inputs $x$ and the transformed inputs $F(x)$ is larger than a threshold value $\tau_1$ (i.e., 0.15), then the regularization item $w_1 \cdot \|F(x) - x\|$ will be added. Note that we calculate input space distance on the preprocessed inputs, and the details of the preprocessing are in § A.2. Following NC [19], the coefficient value $w_1$ is adjusted dynamically to make the reverse-engineering satisfy the constrain (i.e., $\|F(x) - x\| \le \tau_1$). $w_2$ in line 9 and $w_3$ in line 14 are also adjusted dynamically. In lines 8-9, similarly, we add the regularization item for the standard deviation of different Trojan samples' activation values on each pixel in the hyperplane. The default value for $\tau_2$ is 0.25. Lines 10-11 are the standard backward propagation process to update the parameters of the input space transformation function $F$ based on the gradients. The optimizer used to optimize $F$ is Adam [69]. The value of learning rate $lr_1$ is 1e-3. In each epoch, we optimize both the input space transformation function $F$ and the feature space mask $m$.

Lines 12-16 describes the process for optimizing $m$. Similar to line 5, we calculate the cross-entropy loss between the target label and the output of the model under inner features with feature space Trojans in line 12. In lines 13-14, we add the regularization item for the size of the feature space Trojan hyperplane. The default value for $\tau_3$ is 5% of the whole feature space. Lines 15-16 describe the process of updating feature space mask $m$ via gradients. The value of learning rate $lr_2$ in line 16 is 1e-1. The optimizer used is Adam [69].

In line 17, we check if the Trojan is successfully reverse-engineered. In detail, we calculated the ASR (attack success rate) on inner features with feature space Trojans (i.e., $(1 - m) \odot a + m \odot t$). We flag that reverse-engineering is successful if the ASR is above a threshold value $\lambda$ (i.e., 0.8). If the Trojan is successfully reverse-engineered, we flag the model as a Trojan model and label the (source class, target class) pair as Trojaned pair. Besides the details above, we also use K-arm scheduler [21] to speed up the reverse engineering. Lastly, we use Liu et al. [70] to distinguish the Injected Trojans and UAPs (Universal Adversarial Patterns) [71].

## A.2  Details of Datasets

In this section, details of the used datasets are discussed. We also provide the details of the preprocessing for each dataset. All datasets are open-sourced. The license for all datasets is the MIT license. They do not contain any personally identifiable information or offensive content.

**MNIST [56].** This dataset is used for classifying hand-written digits. It contains 60000 training samples in 10 classes. The number of samples in the test set is 10000.

Table 10: Details of Mean and Std value on each dataset.

| Dataset | Mean | Std |
|---|---|---|
| MNIST | [0.1307] | [0.3081] |
| CIFAR-10 | [0.4914, 0.4822, 0.4465] | [0.2023, 0.1994, 0.2010] |
| GTSRB | [0.3403, 0.3121, 0.3214] | [0.2724, 0.2608, 0.2669] |
| ImageNet | [0.4850, 0.4560, 0.4060] | [0.2290, 0.2240, 0.2250] |

**GTSRB [57]** This dataset is built for traffic sign classification tasks. The number of classes is 43. The sample numbers for the training set and test set are 39209 and 12630, respectively.

**CIFAR10 [58]** This dataset is used for recognizing general objects, e.g., dogs, cats, and planes. It has 50000 training samples and 10000 training samples. This dataset has 10 classes.

**ImageNet [59]** This dataset is also a general object classification benchmark. Note that we use a subset (containing 200 classes) of the original ImageNet dataset specified in ISSBA [9]. The subset has 100000 training samples and 10000 test samples.

Following standard convention on the image classification task, we scale the inputs to the range [0,1] and use mean-std normalization to preprocess the images. In detail, the preprocessing can be written as $x' = \frac{(\frac{x}{255} - Mean)}{Std}$, where $x'$ is the normalized input and $x$ is the original inputs. The Mean value and Std (Standard Deviation) value for each channel on different datasets are summarized in Table 10.

## A.3 Details of Attacks

In this section, we discuss the details of the used attacks. By default, the attacks are in all-to-one (i.e., single-target) setting, and the target label is randomly selected when we generate Trojaned models.

**BadNets [1].** This attack uses a fixed pattern (i.e., a patch or a watermark) as Trojan triggers, and it generates Trojan inputs by simply pasting the pre-defined trigger pattern on the input. It compromised the victim models by poisoning the training data (i.e., injecting Trojan samples and modifying their labels to target labels). In our experiments, we use a 3*3 yellow patch located at the left-upper corner as Trojan trigger. The poisoning rate we used is 5%. The attack can be all-to-one (i.e., single-target) and all-to-all (i.e., label-specific). For an all-to-one attack, all Trojan samples have the same target label. For label-specific attacks, the samples in different original classes have different target labels. In our experiment, the target label for label-specific attack is $y_T = \eta(y) = y + 1$, where $\eta$ is a mapping and $y$ is the correct label of the sample.

**Filter Attack [20].** This attack exploits image filters as triggers and creates Trojan samples by applying selected filters on images. Similar to BadNets, the Trojans are injected with poisoning. Following ABS [20], we use a 5% poisoning rate and apply the Nashville filter from Instagram as the Trojan trigger.

**WaNet [7].** This method achieves Trojan attacks via image warping techniques. The trigger transformation of this attack is an elastic warping operation. Different from BadNets and Filter Attack, in this attack, the adversary needs to modify the training process of the victim models to make the attack more resistant to Trojan defenses. It is stealthy to human inspection, and it can also bypass many existing Trojan defense mechanisms [13, 15, 19, 63]. In our experiments, the wrapping strength and the grid size are set to 0.5 and 4, respectively.

**Input-aware Dynamic Attack [8].** This attack generates Trojan triggers via a trained generator network. The trigger generator is trained on a diversity loss so that two different input images do not share the same trigger. Similar to WaNet [7], the attacker needs to control the training process.

**SIG [27].** This method uses superimposed sinusoidal signals as Trojan triggers. In this attack, the attacker can only poison a set of training samples but can not control the full training process. We set the poisoning rate as 5%. The frequency and the magnitude of the backdoor signal in our experiments are 6 and 20, respectively.

**Clean Label Attack [26].** This attack poisons the datasets without manipulating the label of poisoning samples so that the attack is more stealthy. The poisoning samples are generated by a trained GAN. In our experiments, we set the poisoning rate as 5%.

Table 11: Influence of hyperparameters on Trojan mitigation task.

| Metric | $\tau_1$ | | | $\tau_2$ | | | $\tau_3$ | | | | | | |
|---|---|---|---|---|---|---|---|---|---|---|---|---|---|
| | 0.05 | 0.15 | 0.35 | 0.10 | 0.25 | 0.50 | 1% | 2% | 3% | 4% | 5% | 6% | 7% |
| BA | 91.77% | 91.79% | 91.79% | 91.76% | 91.79% | 91.80% | 91.92% | 91.87% | 91.85% | 91.82% | 91.79% | 91.65% | 90.08% |
| ASR | 0.02% | 0.04% | 0.08% | 0.02% | 0.04% | 0.08% | 57.75% | 0.50% | 0.06% | 0.06% | 0.04% | 0.00% | 0.00% |

Table 12: Effects of clean set size on Trojan mitigation task.

| Samples Per Class | BA | ASR |
|---|---|---|
| 5 | 91.03% | 0.08% |
| 10 | 91.79% | 0.04% |
| 50 | 91.66% | 0.02% |
| 100 | 91.66% | 0.06% |

**ISSBA [9].** This attack utilizes an encoder-decoder network to generate sample-specific triggers. The generated triggers are invisible noises. The generated noises also contain the information of a representative string of the target label. The threat model of this attack is that the attacker can only poison the training data, but can not control other components in training (e.g., the loss function). Following the original paper, we poison 10% training data in our experiments.

## A.4 Ablation Study on Trojan Mitigation

In this section, we study the performance of FEATURERE under different constrain values and different numbers of used clean samples. The attack used in this section is WaNet [7].

**Influence of constrain values.** To investigate the influence constrain values (i.e., $\tau_1$, $\tau_2$, and $\tau_3$) on the Trojan mitigation performance, We vary $\tau_1$ from 0.05 to 0.35, change $\tau_2$ from 0.10 to 0.50, and tune $\tau_3$ from 1% of the whole feature space to 7% of the whole feature space. We collect the BA and ASR of the mitigated models and report them in Table 11. The results show that the mitigation performance of FEATURERE is not sensitive to $\tau_1$ and $\tau_2$. For $\tau_3$, when the size of the Trojan hyperplane is extremely small (e.g., 1% of the feature space), the ASR is high. This is understandable because breaking an extremely small feature space Trojan hyperplane means flipping a very small number of neurons, and it is not enough to completely remove the Trojans in the model. Therefore, we set the default value of the hyperplane's size as 5% of the feature space.

**Number of clean reference samples.** To understand the influence of clean set size on the Trojan mitigation task, we vary the number of used clean samples from 5 per class to 100 per class and report the BA and ASR of mitigated model. The results in Table 12 demonstrate that the performance of FEATURERE is robust when the number of used samples changes.

## A.5 Visualization of Reverse-Engineered Trojans

To understand our method and study if it can reverse-engineer Trojans accurately, we visualize the inputs and inner features of clean samples, real Trojan samples, and reversed Trojan samples on nine randomly selected samples in Fig. 4. The model is ResNet18 injected with Filter Trojan [20], Blend Trojan [24] and SIG Trojan [27]. In the feature space, the reverse-engineered Trojan is close to the real Trojan, demonstrating the effectiveness of our reverse-engineering method.

## A.6 Generalization

**Performance on mitigation task for more attacks.** To measure the effectiveness of FEATURERE on Trojan mitigation task, we use more Trojan attacks and report BA and ASR of our method. Besides the results of BadNets [1], Filter [20], WaNet [7] and IA [8] in Table 5, in Table 13, we also show the BA and ASR on LS [1], CL [26] and SIG [27]. The dataset and the model used is CIFAR-10 and ResNet18, respectively. For LS, CL, and SIG, the ASR of FEATURERE is 1.15%, 2.62%, and 1.22%, which are 80.01, 33.18, and 81.22 times lower than that of undefended models. As can be observed, FEATURERE can effectively reduce the ASR while keeping the BA nearly unchanged. Thus, FEATURERE is robust to different attacks on mitigation task.

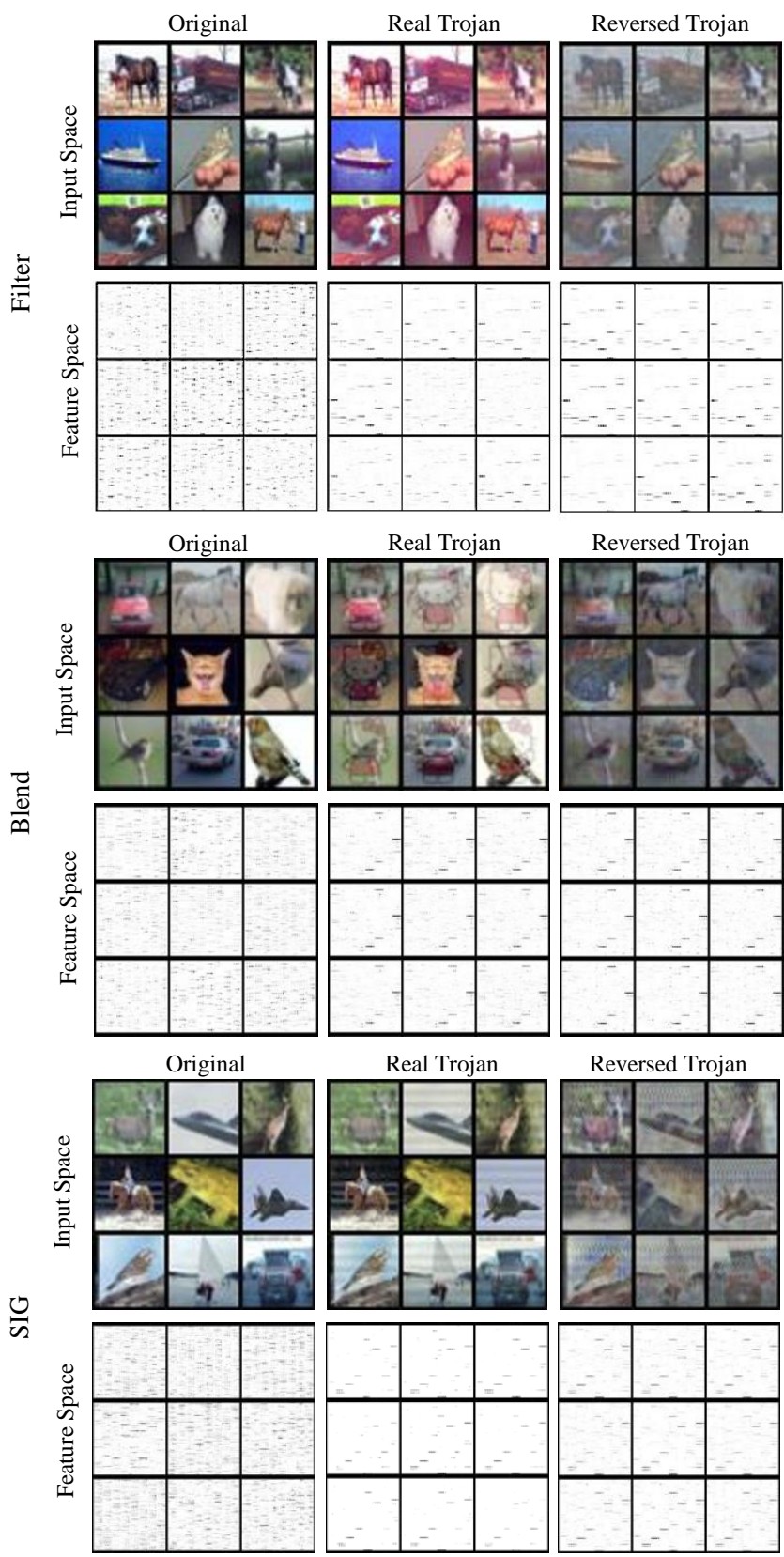

Fig. 4: Visualization of the input space and the feature space for original inputs, real Trojan inputs, and reverse-engineered Trojan inputs.

Table 13: Mitigation results for more attacks.

| Attack | Undefended | | FeatureRE | |
|---|---|---|---|---|
| | BA | ASR | BA | ASR |
| LS | 93.66% | 92.02% | 92.86% | 1.15% |
| CL | 93.51% | 86.94% | 92.94% | 2.62% |
| SIG | 93.73% | 99.09% | 93.47% | 1.22% |

Table 14: Detection accuracy on more models.

| Attack | VGG16 | ResNet18 | PRN18 | LeNet | 4Conv+2FC |
|---|---|---|---|---|---|
| BadNets | 95% | 95% | 100% | 100% | 95% |
| Filter | 90% | 90% | 95% | 90% | 100% |
| WaNet | 90% | 95% | 90% | 90% | 95% |
| IA | 90% | 90% | 90% | 95% | 95% |
| LS | 85% | 90% | 85% | 80% | 85% |
| CL | 80% | 85% | 85% | 80% | 90% |
| SIG | 95% | 95% | 90% | 90% | 90% |

**Generalization to different models.** To understand the generalization of FEATURERE to different model architectures, we evaluate its detection accuracy on BadNets [1], Filter [20], WaNet [7], IA [8], LS [1], CL [26], and SIG [27] attacks using VGG16 [72], ResNet18 [61], Preact-ResNet18 (PRN18) [60], LeNet5 [56], and 4Conv+2FC [51]. The results are summarized in Table 14. In Table 15, we also report FEATURERE's performance on a larger model (i.e., Wide-ResNet34 [73]). In all settings, the detection accuracy is above 80%, and the average detection accuracy on VGG16, ResNet18, and PRN18 is 89.26%, 91.43%, and 90.71%, respectively. FEATURERE achieves high detection accuracy on all different models, demonstrating it is generalizable to different model architectures and larger models.

**Generalization to large input size.** To see if FEATURERE can generalize to large datasets, we report its accuracy on the ImageNette[1] dataset under different attacks. The input size of ImageNette is $3 \times 224 \times 224$. The model architecture used here is Wide-ResNet34 [73]. For each attack, we have 5 Trojaned models. We also train 5 benign models. The results are in Table 15. For all different attacks, the detection accuracy of FEATURERE is above 80%. The average detection accuracy on a large input size is 91.43%. Thus, our method can generalize to large input sizes.

Table 15: Detection accuracy on large input size.

| Attack | TP | FP | FN | TN | Acc |
|---|---|---|---|---|---|
| BadNets | 5 | 0 | 0 | 5 | 100% |
| Filter | 4 | 0 | 1 | 5 | 90% |
| WaNet | 4 | 0 | 1 | 5 | 90% |
| IA | 5 | 0 | 0 | 5 | 100% |
| LS | 3 | 0 | 2 | 5 | 80% |
| CL | 3 | 0 | 2 | 5 | 80% |
| SIG | 5 | 0 | 0 | 5 | 100% |

## A.7 Efficiency

In this section, we measure the efficiency of FEATURERE. Like existing reverse-engineering methods [19, 22, 23], it scans all labels. We optimize this process with a K-arm scheduler [21], which uses the Multi-Arm Bandit to iteratively and stochastically select the most promising labels for optimization. We measure the average runtime on the CIFAR-10 and ImageNet datasets. The model used is ResNet18. The running time on CIFAR-10 and ImageNet are 530.8s and 8934.5s, respectively.

## A.8 Discussions

One finding we have is that using later layers to conduct the reverse-engineering is relatively better than using earlier layers (more results and details can be found in § 4.5). We also found that FEATURERE's performance under the clean-label attack is relatively worse than that of other attacks. We suspect this is because the benign and Trojan features of the clean-label attack are highly mixed. As a consequence, the clean label attack has lower ASR than other attacks. For example, the ASR of the clean-label attack and BadNets are 86.94% and 100.00%, respectively.

---

[1]https://github.com/fastai/imagenette