# OpenReview forum: "Rethinking the Reverse-engineering of Trojan Triggers"
_NeurIPS.cc/2022/Conference — NeurIPS 2022 Accept_

### Official Review · Reviewer_VKAc · 2022-06-29

**Rating:** 6
**Confidence:** 3
**Soundness:** 2 fair
**Presentation:** 2 fair
**Contribution:** 2 fair

**Summary:**

This paper proposes a novel "reverse engineering" method to detect possible Trojans (backdoors) inside models. Unlike previous works mainly aiming to identify input space triggers, this paper considers detecting both input-space and feature-space triggers by formulating the relationships between feature space hyperplane and Trojans. Specifically, the trigger detection problem is formulated as an optimization task on the inner features. This method has been evaluated with state-of-the-art Trojan attacks.

**Questions:**


- Address all questions raised in design and evaluation related comments.

Suggestions for paper improvement:

1. Extend current evaluations to cover all seven attacks and five network architectures, as summarized in the Introduction.
2. Extend evaluations to cover more real-world models (with deep architectures and larger sizes).
3. Evaluate the efficiency of FeatureRE.
4. Refine writing.


**Strengths And Weaknesses:**


Strengths:

- Originality: This paper addresses a previously unsolved problem with unique insight.

- Significance: The proposed method outperforms existing solutions by detecting feature space triggers.

Weakness:

-  Quality: The evaluation seems incomplete and insufficient.

-  Clarity: The method is formally defined. However, some details are not revealed, and the writing is not clear enough.



This paper provides a unique insight that Trojan features will form a hyperplane in the high dimensional space. Based on such insight, the paper extends an input-space trigger detection method (Neural Cleanse) to cover feature-space triggers. The extended solution is formally defined as an optimization task, and the algorithm is clearly listed. However, some details of the solution are not revealed. For instance, this method splits a DNN model into two submodules, A and B (Section 3.3, line 173), but it is not explained how to achieve this. Also, it is not clearly described how a DNN is used to model the function F and why such an approximation works in this optimization task (line 190).


The evaluation part is not satisfactory. The paper states that FeatureRE is evaluated against seven attacks (line 253), but most attacks are not evaluated (in Table 2, Table 3, Table 4, and Table 5). Why not implement all attack methods on all models? Are there any additional difficulties? Table 7 is also confusing. What is the setting for each column? For example, when T1 is, what are the values of T2 and T3?
Another concern is that most of the experiments are conducted on datasets with small inputs. It is not clear how FeatureRE performs on real-world DNN models from perspectives of both speed and accuracy.


Overall, I think this is an interesting work. However, the work needs further improvements, and it is currently not ready for publication.
Other issues
1. line 78: contributions are summered as
2. line 108: Another limitation is they examine
3. line 114: Neural Cleans
4. Eq.1: \mathcal{L} is used but not defined or explained. Also, the attack constraints are unclear in this equation.
5. line 184, line 185, line 196: the statement is vague and unclear.
6. line 213: m is the reversed feature space mask. Is "reversed" redundant?
7. line 230, line 315, line 333: Appendix is mentioned but not attached.

---

> ### Author Response · Authors · 2022-08-02
> **Response to Reviewer VKAc - Part 1**
>
> Thank you for your time and insightful comments. We have run all the suggested
> experiments. We hope the following new clarifications and results can address
> your concerns. We are willing to perform more experiments if you have further
> suggestions.
>
> **Q1:** How to split a DNN model into two submodules, $A$ and $B$?
>
> **A1:** Thanks for your insightful question. The following table shows the
> results of using different $A(x)$ and $B(x)$ on the ResNet18 model and CIFAR-10
> dataset. In detail, we report the results of splitting the model at 9th
> convolutional layer, 11th convolutional layer, 13th convolutional layer, 15th
> convolutional layer, and the last convolutional layer.
>
> Attack | 9-th | 11-th | 13-th | 15-th | Last |
> ---- | ---| ---| --- | --- | --- |
> BadNets | 88% | 93% | 98% |  98% | 98% |
> Filter | 88% | 85%| 90% |  95% | 93% |
> WaNet | 85% | 88%| 85% |   93% | 93% |
> IA | 85% | 83%| 85% |   90% | 95% |
>
>
> In our current implementation, we set $A(x)$ as the sub-model from the input
> layer to the last convolution layer. $B(x)$ is from the last convolution layer
> to the output layer. This is because if the orthogonal phenomenon happens in a
> layer, it will exist for all the subsequent layers. If not, the layer without
> this phenomenon will mix benign and backdoor features, leading to low benign
> accuracy or attack success rate. Thus, a successful backdoor attack will lead
> to the orthogonal phenomenon in the last convolution layer. The above results
> confirm our analysis. We will add a more formal analysis and more results to the
> revised version.
>
>
> ***
>
> **Q2:** How a DNN is used to model the function F? Why such an approximation
> works in this optimization task?
>
> **A2:** Thanks for your valuable question. As we discussed in Line 187-191, we
> use a DNN to represent function F and optimize it via gradient descent.  Our
> goal is to use a DNN to simulate the trigger function, which is equivalent to
> the proposed optimization task. Using DNN to do reverse-engineering is
> popular. Because the DNN has full expressiveness [Hornik et al.], existing
> methods DeepInspect [21] and GanSweep [Zhu et al.] also use similar approaches
> to do reverse-engineering. As pointed out in DeepInspect [21], DNN has full
> ability to learn the probability density distribution of the Trojan trigger.
> We will add more discussions in the revised version.
>
> Zhu et al., GangSweep: Sweep out Neural Backdoors by GAN. ACM MM 2020.
>
> Hornik et al., Multilayer feedforward networks are universal approximators.
> Neural networks 1989.
>
> ***
>
> **Q3:** Most attacks are not evaluated. Why not implement all attack methods
> on all models?
>
> **A3:** Thanks for your insightful comment. Here are the results of the
> required experiments. Please let us know if more is needed.
>
> $\bullet$ Existing results: For the detection task, all seven attacks have
> been covered. Please kindly check Table 6 in this paper. Besides BadNets,
> WaNet, Filter, and Input-aware attacks used in Tables 1-5, the results of the
> label-specific attack (LS), clean-label attack (CL) signal-based attack (SIG),
> and ISSBA can be found in Table 6.
>
> $\bullet$ Results under all attacks on mitigation task: For the mitigation
> task, the results of all attacks can be found in the following table. The used
> network and dataset are ResNet18 and CIFAR-10:
>
> Attack | | Undefended | Ours
> ---- | ---|--- | ---|
> BadNets | BA/ASR (%) | 93.79/100.00  | 93.62/0.74
> Filter | BA/ASR (%) | 91.30/98.98  | 91.26/0.29
> WaNet | BA/ASR (%) | 91.84/98.17  | 91.79/0.04
> IA |BA/ASR (%) | 91.62/92.44  | 91.43/0.38
> LS |BA/ASR (%) |  93.66/92.02  | 92.86/1.15
> CL |BA/ASR (%) |  93.51/86.94  | 92.94/2.62
> SIG |BA/ASR (%) |  93.73/99.09  | 93.47/1.22
>
> $\bullet$ Results under all attacks on different models: We also run all seven
> attacks on all five models. Besides the results of all attacks on the
> ResNet18 model, the accuracy on VGG16, Preact-ResNet18 (PRN18), LeNet5 and 4Conv+2FC with CIFAR-10 dataset is shown in the following
> table:
>
> Attack | VGG16| ResNet18 | PRN18 | LeNet5 | 4Conv+2FC
> ---- | ---|--- | --- |--- | ---
> BadNets | 95% | 95% | 100% | 100% | 95%
> Filter |90% | 90% | 95% | 90% | 100%
> WaNet |90% |  95% | 90% | 90% | 95%
> IA |90% |  90% | 90% | 95% | 95%
> LS |85% |  90% | 85% | 80% | 85%
> CL |80% |  85% | 85% | 80% | 90%
> SIG |95% |  95% | 90% | 90% | 90%
>
> ***
>
> **Q4:** What is the setting for each column in Table 7?
>
> **A4:** Thanks for your helpful comment. The default settings for the
> thresholds are $\tau_1$ = 0.15, $\tau_2$ = 0.25 and $\tau_3$ = 5%. Will
> clarify in the revised version.
>
> ***

---

> > ### Author Response · Authors · 2022-08-02
> > **Response to Reviewer VKAc - Part 2**
> >
> > **Q5:** It is not clear how FeatureRE performs on real-world DNN models from
> > perspectives of both speed and accuracy.
> >
> > **A5:** Thanks for your useful comments.
> >
> > $\bullet$ Accuracy: For the accuracy on the real-world DNN models, we have
> > already reported the performance of FeatureRE on ImageNet dataset with the
> > ResNet18 model in Table 6. The result shows that our method achieves 90%
> > accuracy in this setting.
> >
> > $\bullet$ Efficiency: The running time on ImageNet dataset with the ResNet18
> > model is 8934.5s. The running time on CIFAR-10 and ResNet18 is 530.8s.
> >
> > $\bullet$ More experiments with large input size and deeper architecture: We
> > also conducted more experiments on large input sizes and deeper architecture.
> > In detail, we evaluate FeatureRE's accuracy on ImageNette
> > (https://github.com/fastai/imagenette) dataset with different attacks. The
> > input size for ImageNette is 3x224x224. The model architecture used here is
> > WideResNet34. For each attack, we have 5 trojaned models. We also train 5
> > benign models. The results are shown in the following table, showing that our
> > method has good generalizability on large input sizes and deeper architecture.
> >
> > Attack | TP| FP | FN | TN | Acc
> > ---- | ---|--- | --- |--- | ---
> > BadNets | 5 | 0 | 0 | 5 | 100%
> > Filter | 4 | 0 | 1 | 5 | 90%
> > WaNet | 4 | 0 | 1 | 5 | 90%
> > IA | 5 | 0 | 0 | 5 | 100%
> > LS | 3 | 0 | 2 | 5 | 80%
> > CL | 3 | 0 | 2 | 5 | 80%
> > SIG | 5 | 0 | 0 | 5 | 100%
> >
> > Please let us know if other results are needed. We are happy to perform them
> > and report the results.
> >
> > ***
> >
> > **Q6:** Other minor points.
> >
> > **A6:** Thanks for your valuable comments. We will modify them accordingly.
> >
> > ***

---

> ### Author Response · Authors · 2022-08-09
> **Thanks again for your valuable comments. Does our response address your concerns? We would appreciate the opportunity to engage further if needed.**
>
> Dear reviewer VKAc,
>
> Thanks again for your valuable comments.
> We genuinely hope you could have a look at the new results and clarifications and kindly let us know if they have addressed your concerns.
> We would appreciate the opportunity to engage further if needed.

---

### Official Review · Reviewer_G66q · 2022-07-10

**Rating:** 6
**Confidence:** 4
**Soundness:** 3 good
**Presentation:** 3 good
**Contribution:** 3 good

**Summary:**

Different from existing input-space based methods, this paper designs a novel reverse-engineering method that exploits the feature space constraints to reverse-engineer Trojan triggers. Results on four datasets and seven different attacks demonstrate that the proposed method effectively defends both input-space and feature-space Trojans. It outperforms state-of-the-art reverse-engineering methods and other types of defenses in both Trojaned model detection and mitigation tasks.

**Questions:**

See weaknesses.

**Limitations:**

The authors have addressed the limitations of the work.

**Strengths And Weaknesses:**

Strengths:

1. Good motivation to explore the feature space constraints to reverse-engineer Trojan triggers.

2. The key observation that ‘neuron activation values representing the Trojan behaviors are orthogonal to others’ sounds both interesting and reasonable. And the observation/claim is well supported by Fig.2.
3. Extensive experiments and ablation studies have been conducted to demonstrate the effectiveness of the proposed method.


Weaknesses:

1. This paper assumes that the target label y_t is known, while as I know, Neural Cleanse [17] reverse-engineers a trigger for each label and uses an anomaly detection to identify the potential target label, without knowing the true target label. What if the target label is unknown? How will the problem be formulated differently?

Questions:

1. In Fig.2, it's good to see that the feature spaces of trojan inputs form a linear hyperplane. Will you be able to elaborate what kinds of feature are you using/analyzing? If you use the features/activations of specific layers, what's the creteria to choose the layers? And what's the consequences of using the other layer features?

---

> ### Author Response · Authors · 2022-08-02
> **Response to Reviewer G66q**
>
> Thank you very much for your time and valuable comments. We hope the following
> results and clarifications can address your concerns.
>
> **Q1:** Target label.
>
> **A1:** Thanks for your helpful question. We want to clarify that our method
> does not assume that the target label $y_t$ is known. Like Neural Cleanse, our
> method first scans all labels (Line 2 in Algorithm 1) and identifies the
> potential target labels (Line 17-19 in Algorithm 1). The reverse-engineering
> process starts after identifying the target labels, consistent with the
> existing method such as NC and ABS. We will revise to make it clear.
>
> ***
>
> **Q2:** Will you be able to elaborate what kinds of feature are you
> using/analyzing? If you use the features/activations of specific layers,
> what's the criteria to choose the layers? And what's the consequences of using
> the other layer features?
>
> **A2:** Thanks for your insightful question. We are analyzing the features
> output by the last convolutional layer. The following table shows the results
> of using different $A(x)$ and $B(x)$ on the ResNet18 model and CIFAR-10 dataset.
> In detail, we report the results of splitting the model at 9th convolutional
> layer, 11th convolutional layer, 13th convolutional layer, 15th convolutional
> layer, and the last convolutional layer.
>
> Attack | 9-th | 11-th | 13-th | 15-th | Last |
> ---- | ---| ---| --- | --- | --- |
> BadNets | 88% | 93% | 98% |  98% | 98% |
> Filter | 88% | 85%| 90% |  95% | 93% |
> WaNet | 85% | 88%| 85% |   93% | 93% |
> IA | 85% | 83%| 85% |   90% | 95% |
>
>
> In our current implementation, we set $A(x)$ as the sub-model from the input
> layer to the last convolution layer and $B(x)$ as the rest. This is because if
> the orthogonal phenomenon happens in a layer, it will exist for all the
> subsequent layers. If not, the layer without this phenomenon will mix benign
> and backdoor features, leading to low benign accuracy or attack success rate.
> Thus, a successful backdoor attack will lead to the orthogonal phenomenon in
> the last convolution layer. The above results confirm our analysis. We will
> add a more formal analysis and more results in the revised version.
>
> ***

---

> > ### Comment · Reviewer_G66q · 2022-08-08
> > **Thanks for your response**
> >
> > Dear authors,
> >
> > Thanks for answering my questions. I have no further questions and will maintain my rating.

---

> > > ### Author Response · Authors · 2022-08-09
> > > **Thanks for your feedback**
> > >
> > > Dear Reviewer G66q,
> > >
> > > Thank you very much for your support and feedback. We will make sure the new results and clarifications are properly incorporated into the revised paper.

---

### Official Review · Reviewer_XR7T · 2022-07-12

**Rating:** 7
**Confidence:** 3
**Soundness:** 3 good
**Presentation:** 3 good
**Contribution:** 3 good

**Summary:**

This paper proposes a trojan reverse engineering method by exploiting the feature space input constraint. This is inspired by the authors’ intuition that the trojan and benign features will not affect each other. The paper then validates this intuition by analyzing and visualizing the feature space differences between trojan and benign features. The reverse engineering method is evaluated on popular datasets and models. Results show much better trojan detection performance than prior works. Moreover, the approach can also be used to mitigate trojan attacks and achieves the best mitigation performance.

**Questions:**

- Considering a white-box setting, what the attacker can do to bypass the trojan detection in FeatureRE?

**Limitations:**

See strengths and weaknesses section.

**Strengths And Weaknesses:**

Strengths
- In-depth analysis on neural importance to trojan behavior
- Extensive comparison performed in the evaluation
- Evaluate trojan mitigation effectiveness

Weaknesses
- Lack of adaptive attack evaluation

Overall, this paper presents a very nice insight with validation (although using a quantitative way will be more scientific). The method built upon this insight seems to be sound and has shown much improved performance versus other methods. The only limitation might be that the authors did not evaluate adaptive attacks. Considering a white-box setting, what the attacker can do to bypass the trojan detection in FeatureRE?

---

> ### Author Response · Authors · 2022-08-02
> **Response to Reviewer XR7T**
>
> Thank you very much for your precious time, thoughtful comments, and
> recognition of the novelty and significance of our work. We hope the following
> results and clarifications can adequately address your concerns.
>
> **Q1:** Adaptive attack.
>
> **A1:** Thanks for your valuable comment. Our method is based on the
> observation that the neuron activation values representing the Trojan behavior
> are orthogonal to others. One possible adaptive attack is breaking such
> orthogonal relationships during the Trojan injection process. Based on our
> analysis, we know that this will lead to lower benign accuracy or attack
> success rate.
>
> We design an adaptive attack that adds one loss term to push the Trojan
> features to be not orthogonal to benign features. This attack can be
> formulated as: $L = L_{ce} + L_{adv}$ , where $L_{ce}$ is the standard
> classification loss. Here $L_{adv} = sim(\mathcal{B}(m\odot a + (1-m) \odot
> t), \mathcal{B}(m\odot a^{\prime} + (1-m) \odot t))$. $sim$ is the function to
> measure the cosine similarity. $a$ and $a^{\prime}$ are the features of
> different benign samples. $m$ and $t$ are the feature-space mask and pattern
> of the compromised neurons obtained via SHAP [46] (more details about how to
> get Trojan features and compromised neurons can be found in Section 3.2). The
> loss term $L_{adv}$ can force the Trojan features being not orthogonal to the benign
> ones. We evaluate this adaptive attack on the CIFAR-10 dataset and ResNet18
> model, and the detection accuracy of FeatureRE drops to 0.65. In the meantime,
> the average BA/ASR of the adaptive attack and BadNets (native training) is 87.36%/94.34% and
> 93.67%/99.98%, respectively. As we can see, both the BA and ASR are lower than
> those of native training.
>
> We will add more discussions in our revised version.
>
> ***

---

### Official Review · Reviewer_BzAx · 2022-07-16

**Rating:** 6
**Confidence:** 4
**Soundness:** 2 fair
**Presentation:** 3 good
**Contribution:** 3 good

**Summary:**

*Paper Summary*

Traditionally, most trojan detection methods focus on reverse-engineering a fixed trojan pattern on the input pictures.
 This paper proposes to reverse-engineer the trojan in the feature space of DNNs. The proposed method can be used to detect feature-based trojan or dynamic trojan that is input dependent.

To do so, the author proposes to train a mapping function (F) that maps the original input to the input with the trojan feature.  Also, it trains a mask (M) in the feature space that identifies the trojan features. Various tricks are proposed to train F and M iteratively. The result shows this new method can be used to detect complicated features-based trojan DNNs, such as WaNet.



**Questions:**

*Questions*

1) I am confused about how you know which layer's output is the trojan feature-space. In other words, how many layers are in A(X) and B(X), respectively? Do you need to scan all the feature space from different layers' output?
2) I think you should elaborate more on whether the reverse-engineered trojan feature is interpretable. I noticed that an example is given in the appendix, it is good.  Can your method recover the feature of other trojan methods visually? For example, SIG (strips) or Blended-based attack (hello-kitty pattern).
3) In neural cleanse, the mask is normalized using L1 and in your paper, you switched it to L2. Can you discuss why did you make that decision?

Minor:
In algorithm 1, I cannot find the definition of e/E. Also, are w1-w3 hyperparameters for the loss function? (Usually, people use w to represent weights.) Please also double-check if those notations are defined.


=== Post Rebuttal==
Updated the score to weakly accept.


**Limitations:**

The author does not talk too much about the real limitations.

**Strengths And Weaknesses:**


*Strength*

1) Reverse-engineer the feature of the trojan model is new.
2) This paper provides some good observations of how trojan features manipulate the feature space compared to benign inputs.
3) It outperforms traditional trojan detection methods in both patched-based trojan and feature-based trojan.

*Weakness*
1) The detection of feature-based trojan DNNs, such as WaNet or [22], is a very difficult task. I wish to see more insights regarding this problem. However, the evaluation section fails to give more insight besides the accuracy of the detection.

2) The algorithm is highly tunable with more than 6 hyperparameters. The author does provide a hyper-parameter analysis and shows these hyper-parameters are not effective in the final trojan detection accuracy. However, this is counter-intuitive. For example, if  \tau_3 is not effective, why do you need to train the mask (m) with l2 norm anyway?

3) In my opinion, the novelty of the paper is not very exciting and the training schemes (algorithm 1) are pretty standard. The trojan detection problem is very difficult and the solution seems too simple to me.

---

> ### Author Response · Authors · 2022-08-02
> **Response to Reviewer BzAx - Part 1**
>
> Thank you very much for your time and thoughtful comments. We hope the
> following results and clarifications can adequately address your concerns.
>
> **Q1:** I wish to see more insights regarding this problem. However, the
> evaluation section fails to give more insight besides the accuracy of the
> detection.
>
> **A1:** Thanks for your helpful comments. We had several findings, and we will
> integrate them into our next version. One finding is that using later layers
> to conduct the reverse-engineering is relatively better than using earlier
> layers (more results and details can be found in Q4). We also found that
> FeatureRE's performance under the clean-label attack is relatively worse than
> that of other attacks. We suspect this is because the benign and Trojan
> features of the clean-label attack are highly mixed. As a result, the clean
> label attack has lower ASR than other attacks. For example, the ASR of the
> clean-label attack and BadNets are 86.94% and 100.00% respectively. We will
> extend our analysis and discuss more findings and insights.
>
> ***
>
> **Q2:** The author does provide a hyper-parameter analysis and shows these
> hyper-parameters are not effective in the final trojan detection accuracy.
> However, this is counter-intuitive. For example, if $\tau_3$ is not effective,
> why do you need to train the mask (m) with $\ell_2$ norm anyway?
>
> **A2:** Thank you for the comment. The following table shows the result of
> using more extreme values for these hyper-parameters:
>
> $\tau_1$ | 0.01 | 0.05 | 0.15 | 0.35 | 0.50 |0.75 |
> ---- |---| ---|--- | --- |--- |-- |
> Accuracy| 50% | 95% | 95% | 93% | 58% |50% |
>
> $\tau_2$ | 0.01 | 0.10 | 0.25 | 0.50 | 1.00 | 1.50 |
> ---- |---| ---|--- | --- | --- |--- |
> Accuracy| 50% | 90% | 95% | 93% | 85% |70% |
>
> $\tau_3$ | 0.1% | 3% | 5% | 10% | 50% | 75% |
> ---- |---| ---|--- | --- | --- |--- |
> Accuracy| 50% | 90% | 95% | 93% | 55% |53% |
>
> Our method is robust when the value is in a relatively large region. However,
> if not bounded by $\ell_2$-norm, the results are not good (the accuracy is
> only 50%).
>
>
> ***
>
> **Q3:** The novelty of the paper is not very exciting and the training schemes
> (algorithm 1) are pretty standard. The trojan detection problem is very
> difficult and the solution seems too simple to me.
>
> **A3:** Thanks for your comment.
>
> While existing reverse-engineering methods consider the input-space
> constraints, they fail to reverse-engineer feature-space Trojans whose trigger
> is dynamic in the input space. To the best of our knowledge, we are the first
> to propose feature-space reverse-engineering methods for backdoor detection.
> The method is novel in uncovering feature space constraints and designing the
> feature-space reverse-engineering method.
>
> Reverse-engineering the feature space is challenging and designing Algorithm 1
> is non-trivial. In the input space, all values have natural physical semantics
> and constraints, e.g., a pixel value in the RGB value range. Values in the
> feature space have uninterruptible meanings and are not strictly constrained.
> It is uncertain whether the result will have a physically meaningful semantic.
> In Algorithm 1, we simultaneously optimize the input space trigger function F
> and feature space mask to ensure that the feature space Trojan hyperplane can
> be realized by F on the benign inputs. We will put more discussions in the
> revised version.
>
> ***
>
> **Q4:** I am confused about how you know which layer's output is the trojan
> feature-space. In other words, how many layers are in $A(x)$ and $B(x)$,
> respectively? Do you need to scan all the feature space from different layers'
> output?
>
> **A4:** Thanks for your valuable question. The following table shows the
> results of using different $A(x)$ and $B(x)$ on the ResNet18 model and CIFAR-10
> dataset. In detail, we report the results of splitting the model at 9th
> convolutional layer, 11th convolutional layer, 13th convolutional layer, 15th
> convolutional layer, and the last convolutional layer.
>
> Attack | 9-th | 11-th | 13-th | 15-th | Last |
> ---- | ---| ---| --- | --- | --- |
> BadNets | 88% | 93% | 98% |  98% | 98% |
> Filter | 88% | 85%| 90% |  95% | 93% |
> WaNet | 85% | 88%| 85% |   93% | 93% |
> IA | 85% | 83%| 85% |   90% | 95% |
>
> In our current implementation, we set $A(x)$ as the sub-model from the input
> layer to the last convolution layer. $B(X)$ is from the last convolution layer
> to the output layer. This is because if the orthogonal phenomenon happens in a
> hidden layer, it will exist for all the subsequent layers. If not, the layer
> without this phenomenon will mix benign and backdoor features, leading to
> lower benign accuracy or attack success rate. Thus, a successful backdoor
> attack will lead to the orthogonal phenomenon in the last convolution layer.
> The above results confirm our analysis. We will add a more formal analysis and
> more results in the revised version.
>
> ***

---

> > ### Author Response · Authors · 2022-08-02
> > **Response to Reviewer BzAx - Part 2**
> >
> > **Q5:** Can your method recover the feature of other trojan methods visually?
> >
> > **A5:** Thanks for your insightful question. Yes, our method can recover the
> > feature of other attacks. We will add figures to show the reverse-engineered
> > triggers in our revised version.
> >
> > ***
> >
> > **Q6:** In neural cleanse, the mask is normalized using L1 and in your paper,
> > you switched it to L2. Can you discuss why did you make that decision?
> >
> > **A6:** Thanks for your insightful comment. $\ell_1$ is a better measurement
> > for the image pixel change, and $\ell_2$ is a better measurement for feature
> > space change. Neural Cleanse assumes the backdoor trigger are small patches.
> > Due to the sparse property of $\ell_1$-norm, Neural Cleanse uses it to
> > reverse-engineer the sparse trigger mask, which is close to the small patch
> > trigger.
> >
> > In this paper, we focus on reverse-engineering the feature space trigger that
> > can change all pixels in the image. In such scenarios, $\ell_1$ is not bounded
> > during the attack phase. Thus, recovering sparse trigger masks will lead to
> > inaccurate reverse-engineered triggers. Therefore, we use $\ell_2$-norm
> > following existing works on studying feature space changes (e.g., Xu et al.,
> > AAAI 2021).
> >
> > We also conduct the experiments using $\ell_1$-norm on the CIFAR-10 dataset
> > and ResNet18 model. The results are shown in the following table. It
> > demonstrates that $\ell_1$-norm works well on input-space attack BadNets.
> > However, it has lower performance when facing feature-space attacks (i.e., the
> > Filter attack).
> >
> > Attack | $\ell_1$-norm | $\ell_2$-norm |
> > ---- |--- | --- |
> > BadNets | 98% | 98% |
> > Filter | 85% | 93% |
> >
> > Xu et al., Towards Feature Space Adversarial Attack by Style Perturbation. AAAI 2021.
> >
> > ***
> >
> > **Q7:** Other minor points.
> >
> > **A7:** Thanks for your valuable comments. We will revise them in our revised version.
> >
> > ***

---

> > > ### Comment · Reviewer_BzAx · 2022-08-09
> > > **Response to the authors**
> > >
> > > Thanks so much for the rebuttal.
> > >
> > > (Regarding Q1) These insights are interesting. However, I need more corresponding visualization proofs (as mentioned in Q5).
> > >
> > > (Regarding Q3) I know the trigger is dynamic and this is a complicated problem. The overall solution is not very exciting (nothing beyond my expectation).
> > >
> > > I will raise my score to `weakly accept'. But I will not be upset if the paper gets rejected.

---

> > > > ### Author Response · Authors · 2022-08-09
> > > > **Thanks**
> > > >
> > > > Dear Reviewer BzAx,
> > > >
> > > > Thank you very much for your support and feedback. We will add more corresponding visualization proofs, and make sure the new results are properly incorporated into the revised paper.

---

### Author Response · Authors · 2022-08-02
**Rebuttal Summary**

We sincerely thank all reviewers for their thoughtful comments and precious
time. We provide our responses below to address the concerns. Please let us
know if there is anything still not clear. We are willing to answer more
questions and perform more experiments if the reviewers have further concerns.

---

### Author Response · Authors · 2022-08-08
**Revision Summary**

We thank all reviewers again for their thoughtful questions and suggestions. Below is our revision summary:

**[Introduction, Section 3]** We added clarification for the novelty and the contribution of our method, following the suggestion of Reviewer BzAx.

**[Section 3]** We added clarification that our method does not assume that the target label is known, following the suggestion of Reviewer G66q.

**[Section 3]** We added more discussion about why using DNN to model the function F is effective, following the suggestion of Reviewer VKAc.

**[Section 4]** We added the clarification for the default settings in Table 7, following the suggestion of Reviewer VKAc.

**[Appendix Section A.5]** We added figures of the reverse-engineered Trojan features for more attacks, following the suggestion of Reviewer BzAx.

**[Section 3, Appendix A.6]** We added discussion for how to split a DNN model into two submodules, A and B, following the suggestion of Reviewer BzAx, Reviewer G66q, and Reviewer VKAc.

**[Appendix Section A.7]** We added the discussion about the adaptive attack, following the suggestion of Reviewer XR7T.

**[Appendix Section A.8]** We added the results of implementing all attack methods on different models, following the suggestion of Reviewer VKAc.

**[Appendix Section A.8]** We added the results for generalization to larger models and large input size, following the suggestion of Reviewer VKAc.

**[Appendix Section A.9]** We added the discussion about the efficiency, following the suggestion of Reviewer VKAc.

**[Appendix Section A.10]** We added the discussion about the insights and findings found in the evaluation, following the suggestion of Reviewer BzAx.

**[Other minor issues]** We fixed minor issues, following the suggestion of Reviewer BzAx and Reviewer VKAc.

---

### Meta-Review · Area_Chair_csCz · 2022-08-26

**Recommendation:** Accept
**Confidence:** Certain

**Metareview:**

This paper proposes a new reverse-engineering method for trojan attack detection. The idea is to focus on feature representation space so that the detection is more robust to dynamic / input-dependent attacks and other feature-based attacks. The reviewers consider the idea generally novel and effective, and the experiments thorough. Some reviewers hope to see more visual analysis that can provide better insights into the effectiveness of the method.

**Award:**

No

---

### Decision · Program_Chairs · 2022-09-14

Accept